# Native annual forbs decline in California coastal prairies over 15 years despite grazing

**Josephine C. Lesage** [1,2], **Grey F. Hayes** [3], **Karen D. Holl** [1] *

**1** Environmental Studies Department, University of California Santa Cruz, Santa Cruz, CA, United States of America, **2** Earth and Environmental Science Department, Clark College, Vancouver, WA, United States of America, **3** Swanton Pacific Ranch, Cal Poly San Luis Obispo, Davenport, CA, United States of America

* kholl@ucsc.edu

**Data Availability Statement:** The data are available at https://datadryad.org/stash/dataset/doi:10.7291/D10X19

**Funding:** The original survey was funded by a grant (#99-35101-8234) of the U.S. Department of

## Abstract

Livestock grazing is often used as a land management tool to maximize vegetation diversity in grassland ecosystems worldwide. Prior research has shown that cattle grazing benefits native annual forb species in California's coastal prairies, but drought and increasing aridity may alter this relationship. In 2016 and 2017, we resurveyed the vegetation structure, native annual forb cover, and native annual forb richness in ten grazed and ungrazed prairies that were originally measured in 2000 and 2001 along a 200-km gradient from Monterey to Sonoma counties in California. We found that grazed prairies continued to have significantly lower vegetation height and thatch depth than ungrazed prairies, and that shrub encroachment over the 15-year period was significantly greater in ungrazed prairies. Furthermore, grazed prairies continued to have greater native annual forb richness (4.9 species per site) than ungrazed sites (3.0 species per site), but native annual forb richness declined by 2.8 species per site in grazed prairies and 0.1 species per site in ungrazed prairies between survey periods. We suggest that severe drought and increasing aridity may be driving declines in native annual forb richness in grazed prairies. The species we recorded only in earlier surveys were disproportionately wetland-associated and had higher average specific leaf area than species that remained through the second survey period. Finally, the cover of native annual species increased regardless of whether prairies were grazed, suggesting that the high precipitation in 2017 may have benefitted the native annual forb species that persisted at sites between surveys. Our study shows that weather conditions affect the outcomes of land management strategies.

## Introduction

Most of the Earth's grassland biomes evolved with large ungulate grazing [1], but humans have substantially altered grazing regimes by introducing domesticated ungulate grazers and managing grazing systems across more than a quarter of global land surface [2]. Inappropriate grazing regimes can result in widespread ecosystem degradation and diversity loss, especially in arid and semi-arid environments [3–5]. Well-managed livestock grazing, however, can benefit native vegetation diversity and cover in grass-dominated ecosystems [6–9]. At moderate stocking rates in mesic grasslands, large mammal grazing increases the floral diversity of South

Agriculture (https://nifa.usda.gov/grants) to K.D.H. The resurvey was funded by a small grant from the Santa Clara Valley Chapter of the California Native Plant Society (https://www.cnps-scv.org/education/scholarships/32-scholarships/256-student-research-scholarships) to JCL; funding from the Jean Langenheim graduate fellowship at the University of California, Santa Cruz to JCL; and by the Griswold Endowed Chair funds at UC Santa Cruz to KDH. The funders had no role in study design, data collection and analysis, decision to publish, or preparation of the manuscript.

**Competing interests:** The authors have declared that no competing interests exist.

American steppe [10], Mediterranean-climate grasslands [9, 11], midwestern United States tallgrass prairies [12], and northern European semi-natural grasslands [13]. Furthermore, livestock grazing often reduces the spread of woody shrubs and trees into grasslands, precluding successional conversion to shrubland or forest in the absence of disturbance [14–17], also known as shrub or forest encroachment. However, overgrazing in shrubland ecosystems can facilitate woody encroachment, highlighting the importance of considering the ecosystem type and grazing regime in evaluating the ecological impacts of grazing [2].

In Mediterranean climates, grazing generally benefits short-statured species [18–20], though the strength and direction of this effect can vary depending on annual precipitation [21–23]. In systems like California coastal prairies, where non-native vegetation grows quickly and can limit native species germination and growth, grazing in grasslands has been promoted as a conservation strategy to enhance the diversity of native species [18, 24, 25], as it serves to reduce shrub encroachment and both cover and litter (thatch) of fast-growing, tall-statured exotic annual grasses.

We focus on California's coastal prairies, a highly diverse grassland type. These grasslands are present from Santa Barbara county into Oregon, where winter precipitation is relatively high and coastal fog alleviates summer drought [26]. Among North American grassland ecosystems, coastal prairies are exceptional for their high native species richness and they have higher native species cover than most other grassland types in California [27]. Native annual forbs represent between 25–60% of the recorded species in California's coastal prairies, although they often do not contribute greatly to vegetation cover [18, 26–28].

In 2000 and 2001, Hayes and Holl (18) found strong evidence that moderate levels of cattle grazing favored small-statured native annual forbs in California coastal prairies. In 2016 and 2017, we sought to answer whether such grazing continues to benefit native annual forbs in California's coastal prairies, particularly given the extreme drought and temperature conditions in the years preceding the study. California is already experiencing, 1) progressively more 'extreme' precipitation, wherein rainfall events occur less frequently, but in larger magnitudes; 2) increasing 'whiplash' weather, when exceptionally dry and wet periods follow one another with minimal change in net precipitation; and 3) rising temperatures [29–31]. Over the last few decades, California grasslands have experienced increasing temperatures and aridity [32] and a 1-in-1200-year drought from 2012–2014 [33]. Both these climatic anomalies have impacted the species composition of California grasslands, leading to declines of some drought-sensitive species, particularly native annual forbs [34, 35].

We hypothesized that native annual forb richness and cover had declined in these grasslands since 2001 due to the recent severe drought and overall increasing aridity, but that declines in richness and cover would be less severe in grazed grasslands. To evaluate whether potential declines were related to changes in weather and/or changes in the surrounding vegetation structure, we analyzed climate-relevant traits of the native annual forb species in our sites and changes in vegetation structure over time. We compared the wetland indicator status of species that were recorded in the earlier but not later surveys, which should reflect whether those species were more adapted to mesic environments. We also examined species' specific leaf area (SLA), which is often associated with drought tolerance [35–37]. We hypothesized that the native annual forb species missing in the later survey would largely be adapted to moister conditions.

## Materials and methods

### Site descriptions

In 2000 and 2001, we surveyed 26 paired grazed and ungrazed coastal prairie sites across 400 km of coastline between San Luis Obispo and Mendocino counties in California, USA [18]. In

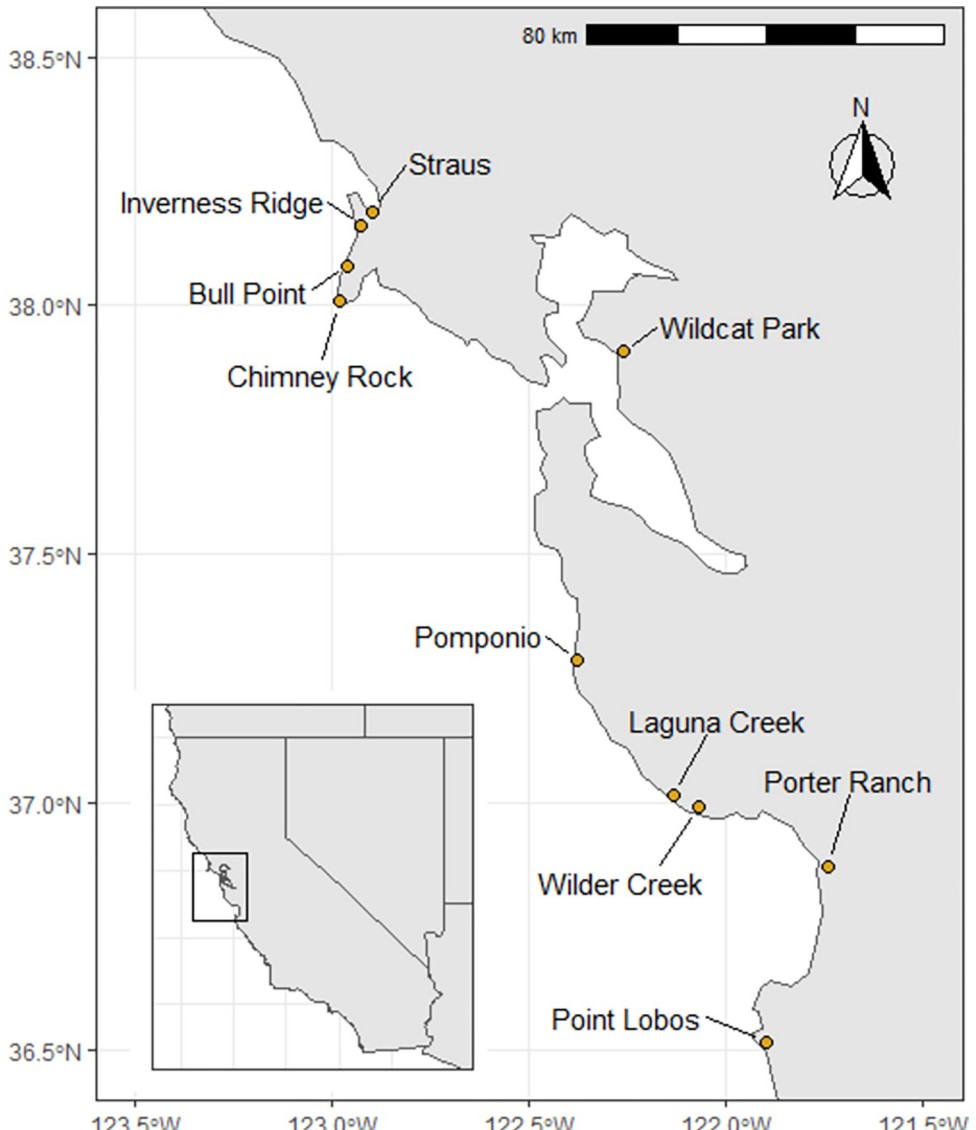

**Fig 1. Locations of sites sampled.** For site details see S1 Table. Data for state perimeters are from the US Department of Commerce, Census Bureau.

2016 and 2017, we resampled sites along 200 km of coastline, in the central area of the prior survey extent (Monterey to Sonoma Counties) where we could obtain permission from landowners for access and grazing patterns had been maintained since 2000–2001 (Fig 1; S1 Table). All analyses presented in this paper include only data from these ten sites, which are a subset of the original 26 sites. For simplicity, we refer to the 2000–2001 sampling period as Time 1 (T1) and the 2016–2017 sampling period at Time 2 (T2). Thus, each sampling period represents two subsequent years of sampling, and all but two sites were sampled four times (S1 Table).

At all sites paired plots are >2 km apart; are on similar slopes and aspects; and range from 1–20 ha and from 20–350 m in elevation. All sites have 1970–2000 30-year temperature and precipitation means within 1.3°C and 544 mm of one another based on interpolated climate data [S1 Table; 38]. There was no evidence of prior mechanical soil disturbance, and soils were deeper than 40 cm and not ultramafic (serpentine).

We compiled available grazing regime information from landowners and agency reports, but in many cases, specific data on the intensity and timing of grazing for our sites were not available. Cattle were the only domestic grazing animals at our sites. The typical grazing regime in the region is at 1 cow-calf pair per 2–4 ha with grazing occurring year-round at most sites by beef or dairying operations. Based on our conversations with landowners, grazing pressure at the second sampling period (T2) was maintained at or slightly below T1 levels. The only wild ungulate at these sites is the Colombian black tailed mule deer (*Odocoileus hemionus columbianus*); we did not find evidence of grazing by this species, but access is not restricted at our sites. Deer primarily eat broadleaved plants and cattle primarily graminoids [39]. Moreover, a recent study suggests that wild ungulate grazing in California has minor impacts on plant communities relative to cattle grazing [40].

## Climate and weather data

California's Mediterranean ecosystems have high interannual variability in rainfall quantity and timing, though all precipitation generally falls between October and April and rainfall between May and September is rare. We gathered all precipitation and temperature information available from weather stations in Santa Cruz, Monterey, San Francisco, and Marin counties from NOAA's Climate Data Online archive (https://www.ncdc.noaa.gov/cdo-web/). Precipitation in the 2000 water year (1 October 1999 to 30 September 2000) was near the 100-year average (449 to 1328 mm) and in 2001 was below average (416 to 912 mm) at our study sites. Precipitation was close to average in the 2016 water year (511 to 1085 mm) and substantially greater than average in 2017 (673 to 1958 mm). Growing season precipitation (November-April) followed similar trends, and temperatures were similar in both sets of sampling years (Fig 2). However, the second set of sampling years occurred immediately after an exceptional 1-in-1200-year drought [33, 41], which was accompanied by above-average temperatures (Fig 2).

## Vegetation structure and native annual forbs

Data collection replicated methods used by Hayes and Holl (2003). We sampled from April through May, traveling from southern sites to northern sites to follow the peak flowering

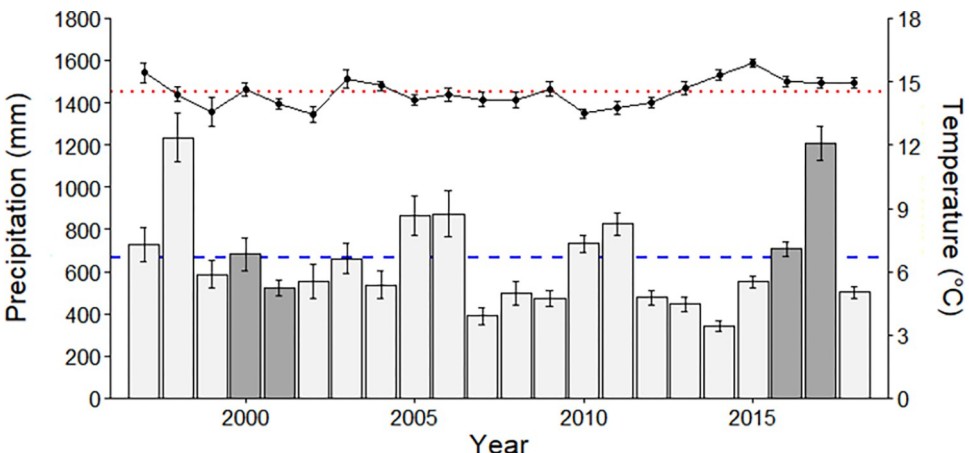

**Fig 2. Mean annual temperature (points) and precipitation (bars) for meteorological stations in the sampling region (including Monterey, Santa Cruz, San Francisco, and Marin counties).** The red dotted line shows the average temperature for the region over this period and the blue dashed line shows mean annual precipitation. Error bars indicate ±1 SE. The sampling years are shaded.

phenology of most species. At each site, we relocated 50-m line transects from the first (T1) sampling period using a Garmin eTrex 20 GPS. There were five transects each in the grazed and ungrazed portions at each site. Transects within sites were separated by 15 m to 1 km and placed at least 5 m from fence lines to avoid edge effects.

We quantified community composition by recording the identity of each species that intersected a 1.8-mm-diameter pin at 1-m intervals. We recorded each species that intercepted the pin once. Our nomenclature and species origin information follow the Jepson Manual and online Jepson eFlora database [42]. We measured vegetation height at 5-m intervals using a piece of paper dropped onto foliage and recording the lowest point. We quantified thatch depth (*i.e.*, build-up of dead plant biomass) by pushing a 1.8-mm pin to the soil surface and measuring the tallest standing dead biomass. We calculated shrub cover as the number of woody shrub pin intercepts as a percent of the 50 intercepts total along the transect. Finally, we carefully searched for and measured the aerial cover (to the nearest cm$^2$) of all native annual forbs present within a 1-m belt transect centered over the 50-m transect.

We compiled specific leaf area (SLA, leaf area/leaf dry mass) for 43 of the 56 native annual forbs species from the TRY Plant Trait Database and data collected by others [35, 43, 44]. We also collected data from species present in and near our sites during spring and summer 2018. Most of these data came from grazed plots, since this is where the species were more prevalent, though we selected individuals that showed no evidence of recent grazing. When data for a single species were available from multiple sources, we averaged the values. High SLA is associated with drought-intolerance and low water use efficiency [45, 46]. We used the United States Army Corps of Engineers National Wetland Plant List to determine the wetland indicator status of all native annual forbs found in our sites [47].

### Data analysis

We conducted all data analyses in R version 4.1.2 [48]. To model the effects of time and grazing treatment on vegetation structure (*i.e.*, vegetation height, thatch depth, and shrub cover) and native annual forbs (*i.e.*, richness and cover), we used generalized linear mixed models in the glmmTMB package [49]. We treated sampling period and grazing treatment (both categorical with two levels) as fixed effects and treated year and transect as nested within-site random factors. We fit vegetation and thatch height with normal distributions, shrub cover using a beta distribution, and native richness using a Poisson distribution. Cover data were fit using a Tweedie distribution [50] because our data were overdispersed and zero-inflated. Model fit was checked using the DHARMa package to plot residuals against fitted values [51]. We summarized the SLA of native annual forb species for the sampling period when they were found (*i.e.*: lost, observed in only T1; still present, observed in both T1 and T2; and new, observed in only T2) and compared the values using one-way ANOVA.

### Results

Vegetation height was greater in both ungrazed and grazed prairies in the second sampling period than in the first, and, as expected, vegetation was consistently shorter in grazed than ungrazed prairies across both periods (Fig 3A; Table 1). Grazed prairies had lower thatch depth than ungrazed prairies across all sampling years, and thatch depth did not change significantly over time (Fig 3B; Table 1). Shrub cover increased significantly more in ungrazed prairies than in grazed prairies between the two sampling periods (Fig 3C; Table 1). More than 95% of total shrub cover was comprised of three native species *Baccharis pilularis* (66.4%), *Rubus ursinus* (26.5%), and *Toxicodendron diversilobum* (2.8%).

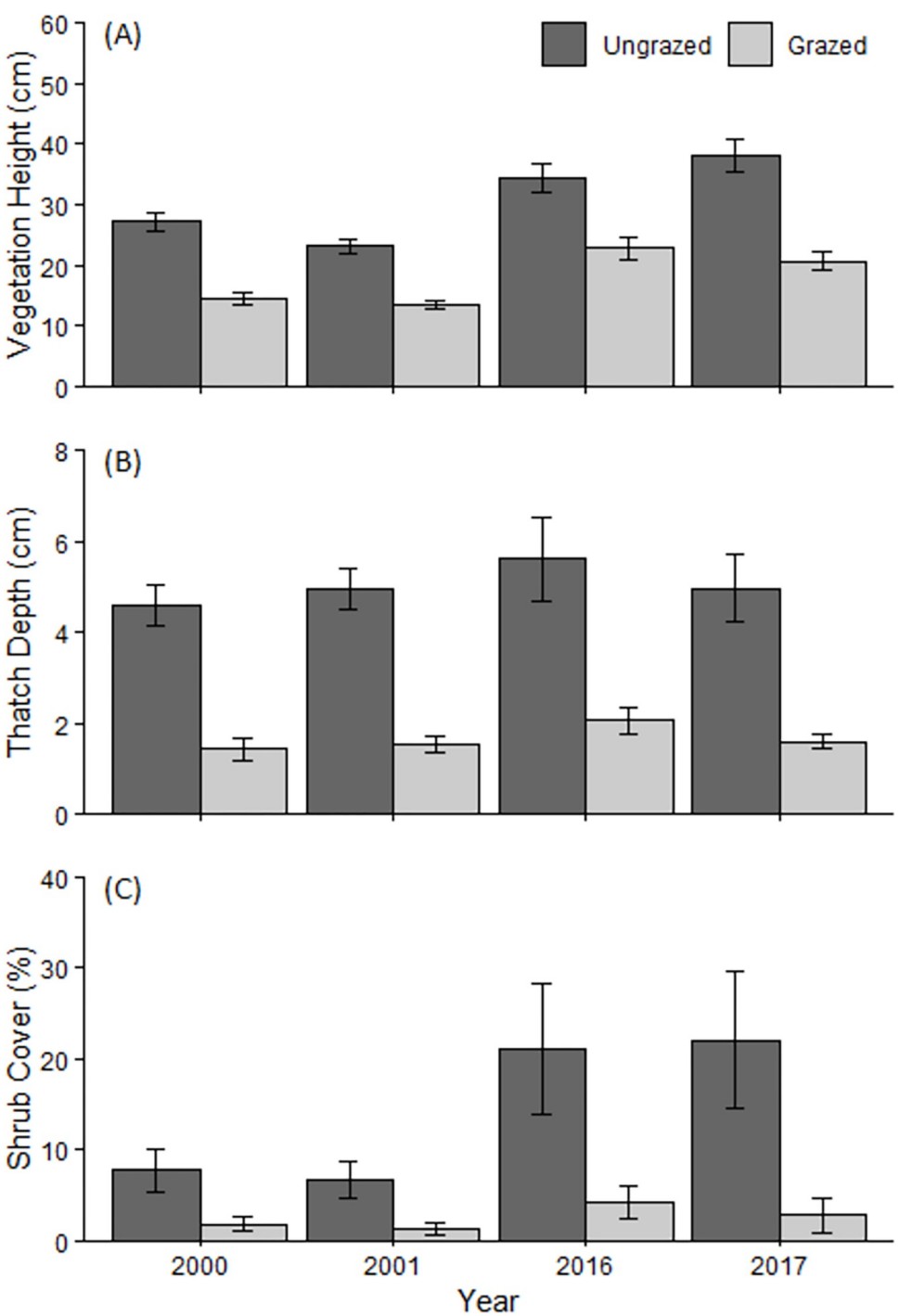

**Fig 3. Vegetation height, thatch depth, and shrub cover over time in grazed and ungrazed prairies.** Values represent means for n = 10 sites and error bar indicate 1 SE.

We recorded a total of 56 annual forb species in grazed and ungrazed prairies during the four survey years (S2 Table). The mean richness of native annual forbs declined between T1 and T2 by 2.8 species in grazed prairies and 0.1 species in ungrazed prairies, resulting in a significant interactive effect of time and grazing treatment on the richness of native annual forbs

**Table 1. Generalized linear model results for changes in vegetation factors across treatments and over time.**

| Model | Parameter | Estimate | Std. Error | z-value | p-value |
|---|---|---|---|---|---|
| Vegetation height | Intercept | 25.87 | 2.44 | 10.60 | <0.001 |
|  | Sampling Period | 10.43 | 1.93 | 5.42 | <0.001 |
|  | Treatment | -11.24 | 1.56 | -7.21 | <0.001 |
|  | Sampling Period × Treatment | -3.47 | 1.96 | -1.77 | 0.077 |
| Thatch depth | Intercept | 4.84 | 0.72 | 6.76 | <0.001 |
|  | Sampling Period | 0.49 | 0.34 | 1.44 | 0.151 |
|  | Treatment | -3.31 | 0.46 | -7.24 | <0.001 |
|  | Sampling Period × Treatment | -0.17 | 0.44 | -0.38 | 0.707 |
| Shrub cover | Intercept | -3.03 | 0.45 | -6.79 | <0.001 |
|  | Sampling Period | 1.24 | 0.22 | 5.52 | <0.001 |
|  | Treatment | -1.14 | 0.27 | -4.18 | <0.001 |
|  | Sampling Period × Treatment | -0.85 | 0.34 | -2.47 | 0.014 |

(Fig 4A; Table 2). Native annual forb richness remained higher in grazed than ungrazed sites in T2, but the difference was much smaller than in T1. Native annual forb cover was greater in grazed than in ungrazed prairies during T1 (Fig 4B; Table 2). During T2, mean native annual forb cover in grazed sites was more than twice as high in 2017 as compared to 2016, but was similar in both years in ungrazed sites.

The average SLA of species observed only in T1 ($342.7 \pm 43.0$ mm$^2$ g$^{-1}$) was greater than those found in both ($227.3 \pm 18.0$ mm$^2$ g$^{-1}$) periods or only in T2 ($168.9 \pm 17.0$ mm$^2$ g$^{-1}$; n = 43 species, F = 3.3, p = 0.020). Most species observed in both sampling periods are classified as upland or facultative upland species (Fig 5). The proportion of species classified as 'obligate wetland', 'facultative wetland', or 'facultative' by the US Army Corps of Engineers declined from T1 (30%) to T2 (23%) and the proportion of facultative upland and upland species increased (Fig 5).

## Discussion

### Grassland structure

The effects of grazing on vegetation structure (i.e., height, cover) were similar over time. In both sampling periods, grazed prairies had shorter standing living canopies and reduced thatch, as expected. The vegetation was significantly taller in both grazed and ungrazed plots in the second than in the first sampling period (T1:19.5 cm; T2: 29.0 cm, Table 1), though the net difference in vegetation height between grazed and ungrazed plots was consistent over time (T1: Δ11.2 cm; T2: Δ14.6). Mean vegetation height in grazed prairies during the second sampling period approached but did not quite reach the T1 height of ungrazed prairies (Fig 3). The greater vegetation height in T2 could be attributed either to more precipitation in T2 than T1 (Fig 2) or lower grazing pressure. Although we cannot conclusively distinguish between these two hypotheses, the fact that vegetation height was similar in both years of each sampling period, despite interannual differences in precipitation, suggests that grazing pressure was lower during T2. This trend towards reduced grazing intensity concurred with anecdotal information from landowners.

Our resurvey supports the use of cattle grazing to reduce the rate of shrub encroachment into California coastal prairies. Many native coastal prairie species are adapted to disturbance regimes that prevent shrub encroachment, having evolved under grazing by Pleistocene

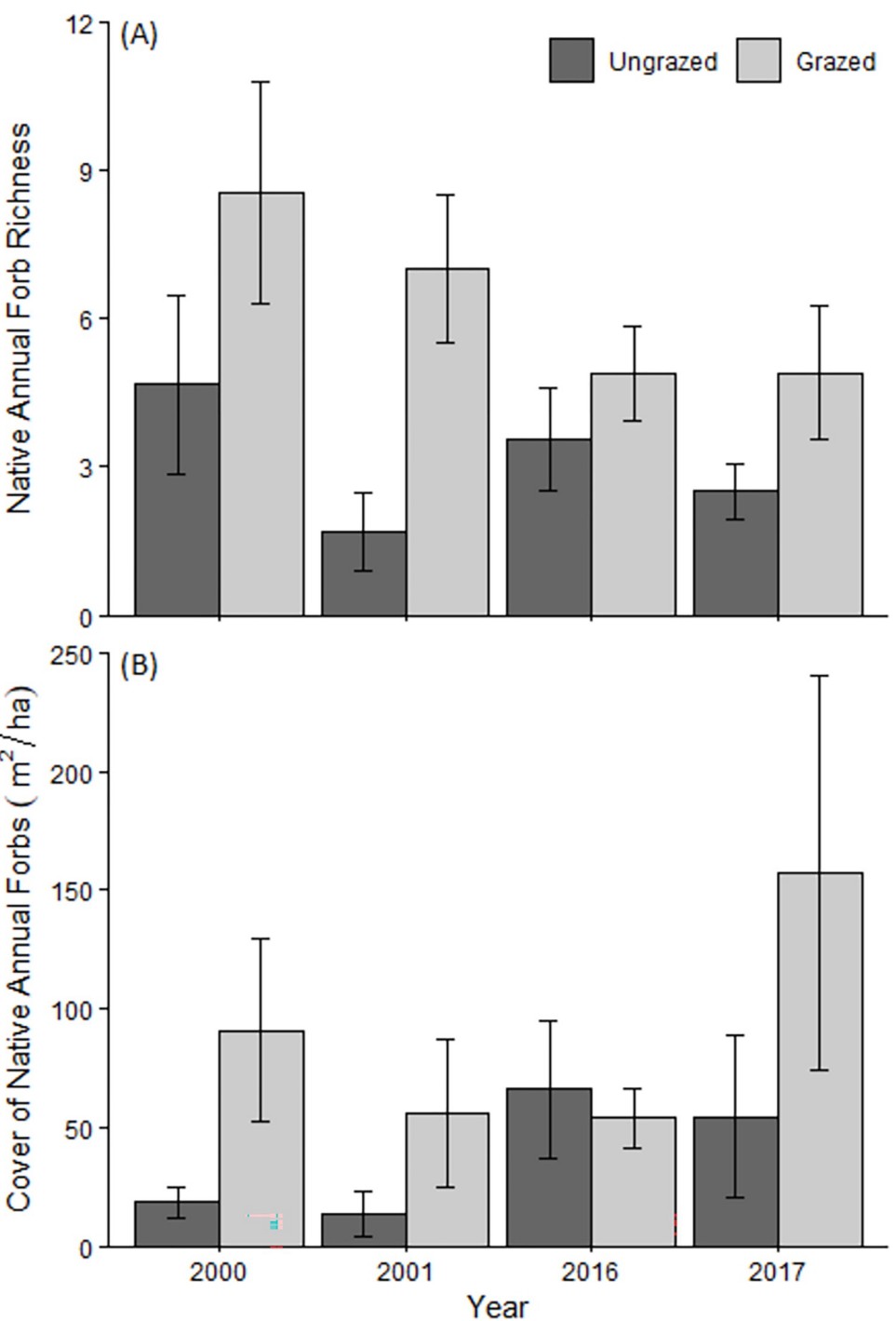

**Fig 4. Richness and cover of native annual forbs in grazed and ungrazed prairies in four survey years (n = 9 in 2000, 2016, n = 10 in 2001, 2017).** Error bars indicate 1 SE.

megafauna [52] and later, frequent burning by Native peoples [53]. Shrub cover increased substantially more in ungrazed than grazed prairies over a 15-year period. The most common shrub species are mostly native, and their rate of spread is similar to those described from other coastal California sites [25, 54].

**Table 2. Fitted models for the effects of grazing and year on native annual forb richness and cover.**

| Model | Parameter | Estimate | Std. Error | z-value | p-value |
|---|---|---:|---:|---:|---:|
| Native annual forb richness | Intercept | -1.10 | 0.37 | -3.01 | 0.003 |
| | Sampling Period | -0.02 | 0.14 | -0.14 | 0.892 |
| | Treatment | 1.09 | 0.16 | 7.01 | <0.001 |
| | Sampling Period × Treatment | -0.39 | 0.16 | -2.40 | 0.017 |
| Native annual forb cover | Intercept | -0.41 | 0.50 | -0.82 | 0.411 |
| | Sampling Period | 1.16 | 0.41 | 2.83 | 0.005 |
| | Treatment | 1.60 | 0.45 | 3.57 | <0.001 |
| | Sampling Period × Treatment | -0.73 | 0.48 | -1.51 | 0.132 |

## Native forb richness and cover

Native annual forb species richness and cover continued to be greater in grazed than ungrazed prairies in the second sampling period, although the richness of grazed prairies had declined substantially over time. Our results are consistent with past research showing that shorter vegetation and higher levels of light at the soil surface increase native annual forb germination and growth when competition with exotic annual grasses is high [55, 56]. Reducing the canopy height and dominance of exotic annual grasses through grazing is a common grassland management technique, as exotic species often have faster growth rates and are competitively dominant to many native species in their early stages, reducing native species establishment [57–60]. Furthermore, the reduction of thatch by grazing can substantially enhance recruitment of native grassland species [61, 62].

The significant decline in the richness of native annual species we measured in grazed prairies complicates the interpretation of prior work showing a beneficial effect of grazing on native annual forbs in California [18, 19] and other Mediterranean grasslands [63–65]. Native annual forb species richness declined significantly more in grazed than ungrazed prairies. This is likely because ungrazed prairies had much lower richness at the first sampling period, comprised of those species best able to compete with tall-statured exotic grasses; consequently, there were fewer species that could disappear during the 15 years between sampling periods. We cannot directly attribute the decline in native annual forb richness to changes in grazing regime or climate, but we use circumstantial evidence to explore the relative importance of these factors.

The significantly taller vegetation height in the second sampling period may explain part of the decline in native annual forb richness, given that many of the native annual forbs in California are low-stature species that benefit from greater light availability afforded by a shorter

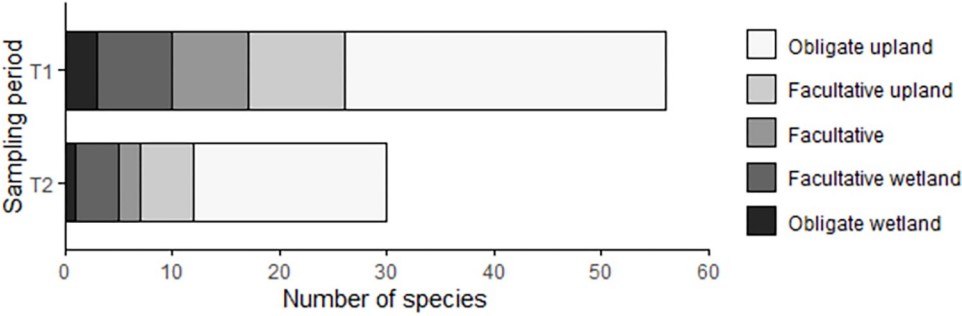

**Fig 5. Wetland indicator status of native annual forbs observed in T1 (2000–2001) and in T2 (2016–2017).**

canopy [18, 19, 60]. However, vegetation height increased in both grazed and ungrazed plots, whereas native annual forb richness only declined in grazed sites. Thatch depth did not differ in grazed plots between the sampling periods and therefore likely did not cause the declines. There was significant treatment × sampling period interaction, as shrub cover increased substantially between sampling periods in ungrazed plots (~7 to 29%), but not in grazed plots (from ~1 to 4%). Thus, it seems unlikely that the primary cause for decline in native annual forb richness in grazed plots was local changes in vegetation structure.

Multiple lines of evidence support the hypothesis that climate and weather factors prior to the second survey led to the decline in native annual forb richness, though we note that we did not conduct manipulative experiments to directly link specific climate variables and species composition. California experienced a severe, 1-in-1200 year drought [33] between our sampling periods and the years immediately preceding our second sampling period were both exceptionally dry and warm (Fig 2). Evidence from wetland indicator status and SLA, both of which are tied to plant water stress tolerance, suggest that increasing aridity, and therefore less plant-available water, may be an important contributor in the richness declines we measured. The native annual forb species absent in the second sampling period were disproportionately adapted to mesic environments according to their wetland indicator status (Fig 5). Moreover, the average SLA of plant species found only in T1 was significantly greater than for those found in T2 or both sampling periods, suggesting that the species that persisted are better adapted to drought. Likewise, other long-term grassland data in California [66] and experimental work by LaForgia and colleagues [67] show that native annual grassland forbs with traits associated with higher moisture conditions are more negatively affected by drought.

Importantly, grazing and precipitation often have interactive effects on native species richness and cover [40, 62, 68, 69]. Species richness generally increases under grazing in mesic and highly productive environments, but richness often declines under grazing in arid environments [10, 19, 70, 71] [but see 40]. Grazing during the extreme drought of 2012–2014 may have directly reduced native annual forb species richness, consistent with prior studies showing that grazing can reduce biodiversity during drought conditions [68, 69].

Overall native annual forb cover was greater in T2 than T1, and native annual forb cover was greater in grazed prairies than ungrazed prairies in three of four survey years (Fig 4). The greater native annual forb cover in ungrazed prairies in T2 as compared to T1 may be attributable to lagged drought-induced declines in exotic annual grasses [46, 72], reducing competitive pressure and allowing some species of native forbs to increase significantly in cover. Furthermore, the large spike in native annual forb cover in some grazed sites in 2017 as compared to 2016, suggests that the higher than typical precipitation in 2017 benefitted the native annual species that persisted through the drought but did little to restore native species richness, reinforcing findings from other sites that 'drought plus deluge do not equal normal' [35].

Our results show a pattern of declining native annual forb richness, but we are unable to determine whether the declines we observed are a short-term response to the 2012–2014 drought, changes in grazing, or a sign of longer-term trends. Heavy winter rainfall in 2017 did not result in increased site-level native annual forb species richness, which would be expected if high precipitation stimulated the persistent seedbanks typical of many species in this group, though a single high precipitation year may not be enough for all species to recover following severe drought. Prior work on the dynamics of annual forbs and annual grasses suggests that annual forbs persist alongside exotic annual grasses in part due to precipitation variability and drought [73, 74]. Additional data are necessary to determine whether the declines in richness we observed were due to local extinction or were only temporary responses to drought.

In conclusion, we find that grazed grasslands continue to have a greater richness of native annual forbs than ungrazed grasslands, but that the gap between grazed and ungrazed prairie

diversity declined. Our study provides additional evidence that drought and rising temperatures affect community composition in managed grasslands globally [75, 76]. Finally, our study raises the question of whether species adapted to mesic conditions need alternative conservation strategies beyond *in situ* grazing in areas where aridity is increasing, such as assisted migration to climate refugia or *ex situ* collections.

## Supporting information

**S1 Table. Sites visited by sampling year, ordered from North to South.** Elevation extracted from the USGS National Map Elevation Point Query Service. Thirty-year mean temperatures and precipitation values extracted from the WorldClim version 2.1 climate model (Fick & Hijmans 2017).
(PDF)

**S2 Table. Native annual forb species observed in sampling periods 1 and 2.** Nomenclature follows Jepson Flora Project (2020).
(PDF)

## Acknowledgments

We thank the UC Santa Cruz Reserve System, California State Parks, National Parks Service, Monterey Peninsula Regional Parks District, and private landowners for allowing us to access their land. We thank S. Glascock and M. Voce for their help in the field. We appreciate helpful feedback from A. Huertas-Herrera, M. Loik, and S. Harrison on drafts of this paper.

## Author Contributions

**Conceptualization:** Josephine C. Lesage, Grey F. Hayes, Karen D. Holl.

**Data curation:** Josephine C. Lesage, Grey F. Hayes.

**Formal analysis:** Josephine C. Lesage.

**Funding acquisition:** Josephine C. Lesage, Karen D. Holl.

**Investigation:** Josephine C. Lesage, Grey F. Hayes.

**Methodology:** Grey F. Hayes, Karen D. Holl.

**Project administration:** Josephine C. Lesage.

**Supervision:** Karen D. Holl.

**Visualization:** Josephine C. Lesage.

**Writing – original draft:** Josephine C. Lesage, Karen D. Holl.

**Writing – review & editing:** Josephine C. Lesage, Grey F. Hayes, Karen D. Holl.

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
