## [Decision Letter · Decision Letter 0]

21 Apr 2022

PONE-D-22-07590Native annual forbs decline in California coastal prairies over 15 years despite grazingPLOS ONE

Dear Dr. Holl,

Thank you for submitting your manuscript to PLOS ONE. After careful consideration, we feel that it has merit but does not fully meet PLOS ONE’s publication criteria as it currently stands. Therefore, we invite you to submit a revised version of the manuscript that addresses the points raised during the review process.

We look forward to receiving your revised manuscript.

Kind regards,

Fei Xu, Ph.D.

Academic Editor

PLOS ONE

Journal Requirements:

2. We noted in your submission details that a portion of your manuscript may have been presented or published elsewhere. [As is clearly stated in this paper, the data are a follow up on a survey conducted 15 years earlier to document changes over time.] Please clarify whether this conference proceeding or publication was peer-reviewed and formally published. If this work was previously peer-reviewed and published, in the cover letter please provide the reason that this work does not constitute dual publication and should be included in the current manuscript.

4. We note that Table Sites-S1 in your submission contain map/satellite images which may be copyrighted. All PLOS content is published under the Creative Commons Attribution License (CC BY 4.0), which means that the manuscript, images, and Supporting Information files will be freely available online, and any third party is permitted to access, download, copy, distribute, and use these materials in any way, even commercially, with proper attribution. For these reasons, we cannot publish previously copyrighted maps or satellite images created using proprietary data, such as Google software (Google Maps, Street View, and Earth). For more information, see our copyright guidelines: http://journals.plos.org/plosone/s/licenses-and-copyright.

a)  You may seek permission from the original copyright holder of Table Sites-S1 to publish the content specifically under the CC BY 4.0 license.  

Reviewers' comments:

Reviewer's Responses to Questions

**Comments to the Author**

1. Is the manuscript technically sound, and do the data support the conclusions?

Reviewer #1: Partly

Reviewer #2: Yes

2. Has the statistical analysis been performed appropriately and rigorously? 

Reviewer #1: Yes

Reviewer #2: Yes

3. Have the authors made all data underlying the findings in their manuscript fully available?

Reviewer #1: Yes

Reviewer #2: Yes

4. Is the manuscript presented in an intelligible fashion and written in standard English?

Reviewer #1: Yes

Reviewer #2: Yes

5. Review Comments to the Author

Reviewer #1: Authors examined the effects of livestock grazing by cattle on grassland diversity and cover in California. This is a timely topic and adapting grazing management to account for changing climate will be critical for conserving rangelands worldwide. The authors are lacking key pieces of information in the methods section, and the discussion section must draw conclusions more tightly linked to actual findings, and contextualize findings in the context of current literature.

Introduction

Page 3, Line 25: I’m not familiar with the term semi-natural plant community, define?

Page 3, Line 38: Is grazing only beneficial in highly invaded grasslands?

Materials and methods

Page 5: no need to include ‘the two of us’

Page 5: Please provide more information about the grazed and ungrazed plots. At what intensity (AUM) were the plots grazed? What size are the plots? When surveying the vegetation did you avoid the fenceline / edge effects? At what frequency were the plots grazed? How close were the paired sites (are they bisected fields?)? Please discuss herbivory within both grazed and ungrazed plots by wild ungulates.

Page 5: Site descriptions. No need to talk about survey permission, just describe the sites that you actually included in the study. Could you include a map of these study areas and/or a description of the range of conditions across which these plots occur (distance apart, elevational range, etc.). In other words, tailor site description to the actual study sites.

I see here that it was hard to establish a complete grazing history, but could you provide a ball park estimate of grazing intensity in order to contextualize that patterns that you observe with other studies?

Line 99: It is going to be hard to attribute differences in response to grazing to climate (especially with incomplete grazing history), with only two time points (though I agree it is likely). We two sample points, however, many other factors could have changed – for instance, land use in the surrounding habitat matrix, nitrogen deposition, pressure from undomesticated livestock. In other words, attributing differences to drought specifically is challenging, and should be discussed in the discussion.

Line 106: Does this sampling regime recapitulate the original veg sampling?

Line 125: Just the version of R is fine – though I do like ‘Bird Hippie’.

Line 128: How is year different from time period – somehow I didn’t understand that you had multiple years of data within each time period – could you make this more clear?

Results

In terms of vegetation height, did you derive this via the trait database or did you collect measurements in the field, and if the latter, were measurements collected after the fields had been grazed? Sorry, that is more of a methods question!

Could you provide more information about the overall community composition – in particular, I’m curious about invasive species cover, perennial grass cover, etc.

Discussion

Line 185: As mentioned, I need a little more information before I can contextualize this height difference.

Line 214: Can you connect your findings here with other studies of grazing in similar habitats?

Line 227: Since your statistical tests indicate no differences, do not discuss as if there are treatment effects.

Line 222: Did changes occur due to composition change or due to increased biomass?

Line 245: This is good – could mention other forms of change in the region that could explain differences, but supporting your finding with the mesic index etc is good.

Line 252: Be clear here about what grazing effects (positive effects on diversity?). Several studies have shown that negative grazing impacts on diversity and native cover are exacerbated during drought years.

Live 253: This transition to annual dominated states has been observed in other studies – maybe cite a few?

Live 276: I don’t feel like you’ve set us up to make conclusions about microhabitat (did you do any spatial studies?) or to mention assisted migration. In order to mention these things, please explain how your findings support this suggestion.

Reviewer #2: Review of PLOS

This is an excellent paper that resamples a coastal prairies to explore how a regional drought influenced species richness. The sampling approach was straightforward, as was the data analysis. The results were unambiguous and certainly worthy of publication. The writing was clear and concise and addressed the key limitations and strengths of the analysis. Overall, a very clean manuscript.

There are a few areas where I believe there are opportunities for improvement or increased impact for this publication. Here are my suggestions:

1. Given the broad regional sampling that was conducted, there may be opportunities to get a more nuanced view of the impact of climate change/drought. It is highly unlikely that the extent of the drought (deviation from mean conditions) or the 2017 pulse in increased precipitation was equal across the entire gradient. I would appreciate seeing the nuance of species loss or increase in vegetation height relative to the extent of drought/2017 pulse across the gradient rather than a single, aggregated measure.

2. Somewhere either in the supplementary materials or in a table in the paper there should be a list of species, their abundance, and an identification of which species were lost. This would be a valuable resource.

A few other minor edits.

Line 40: Provide spatial description for readers not familiar with California geography; consider including biophysical data that capture the environmental gradient.

Line 53: Global patterns don’t help in interpreting these data; need to include weather trends and key disruptive weather events that are useful in understanding the changes observed.

Line 98: More precision here. Much of the explanation hangs on a single source describing an “exceptional drought”. I want to see the actual deviation, including the temporal extent of dry periods, the relative change in water year, increases in mean temperature, etc. for the sampled region.

Line 195: Shrub encroachment as a topic comes out of nowhere. It hasn’t been established as a key metric nor are data presented. I’d drop this argument or add the woody plant data.

Line 221: The most parsimonious explanation is that a higher-than-average water year produced greater plant biomass in this water-limited ecosystem.

Line 237: Unclear what “Climatic Water Deficit” is as a proper noun. First time it is introduced and it isn’t well explained.

Overall, I believe this paper to be valuable to understanding the factorial influence of grazing and weather on forb species richness in a biodiverse ecosystem.

6. PLOS authors have the option to publish the peer review history of their article (what does this mean?). If published, this will include your full peer review and any attached files.

Reviewer #1: No

Reviewer #2: No

---

## [Author Response · Author response to Decision Letter 0]

2 Jun 2022

Please see attached file with responses to each comment in red italicized text.

---

## [Decision Letter · Decision Letter 1]

13 Sep 2022

PONE-D-22-07590R1Native annual forbs decline in California coastal prairies over 15 years despite grazingPLOS ONE

Dear Dr. Holl,

Thank you for submitting your manuscript to PLOS ONE. After careful consideration, we feel that it has merit but does not fully meet PLOS ONE’s publication criteria as it currently stands. Therefore, we invite you to submit a revised version of the manuscript that addresses the points raised during the review process.

We look forward to receiving your revised manuscript.

Kind regards,

Mehdi Heydari

Academic Editor

PLOS ONE

Journal Requirements:

Additional Editor Comments:

Dear Dr. Karen D. Holl

Based on the reviewers comments and my additional control, I am pleased to inform that your manuscript PONE-D-22-07590R1" Native annual forbs decline in California coastal prairies over 15 years despite grazing" can be accepted for publication in PLOS ONE.after minor revision

Reviewers' comments:

Reviewer's Responses to Questions

**Comments to the Author**

1. If the authors have adequately addressed your comments raised in a previous round of review and you feel that this manuscript is now acceptable for publication, you may indicate that here to bypass the “Comments to the Author” section, enter your conflict of interest statement in the “Confidential to Editor” section, and submit your "Accept" recommendation.

Reviewer #3: All comments have been addressed

Reviewer #4: (No Response)

2. Is the manuscript technically sound, and do the data support the conclusions?

Reviewer #3: Partly

Reviewer #4: Yes

3. Has the statistical analysis been performed appropriately and rigorously? 

Reviewer #3: Yes

Reviewer #4: Yes

4. Have the authors made all data underlying the findings in their manuscript fully available?

Reviewer #3: Yes

Reviewer #4: Yes

5. Is the manuscript presented in an intelligible fashion and written in standard English?

Reviewer #3: Yes

Reviewer #4: Yes

6. Review Comments to the Author

Reviewer #3: Dear Authors,

Overall the manuscript is a very nice attempt to explore the response of coastal prairies to changes in grazing management (grazed and ungrazed) in California. The paper is well written, and I think it is pleasant to read. However, some details can be improved, e.g., some unsupported sentences could be deleted, several concepts must be standardized, and minor changes in some figures should be made. Note that I do not find the "wetland indicator status" analyses helpful. But, it is up to you keeping or remove it. Beside this, I believe both supplementary figures should be in the manuscript's main text and not as an annexe. In the S1 Figure, please remove the words Longitude and Latitude; they are unnecessary.

I enclose my specific comments in the Word file.

Bests

Reviewer #4: I have reviewed article 7590R1. I was not a reviewer of the original submission and consequently cannot comment on changes that were made to the previous version. Following my read of this version I have no serious concerns about the science contained in this work nor its interpretation. Some minor issues or suggestion for clarification include:

Line 24 – Specify ‘Livestock grazing’ grazing by native animals is typical of all grasslands etc.

Line 32-34 – There are just as many examples of livestock grazing encouraging woody plant encroachment as there are of it discouraging it. I would temper this statement and include the counterpoint that livestock can encourage encroachment.

Line 79 – Emphasize that these 10 sites are a subset of the original 26

Line 84 – how many transects? I believe this question is answered later, but transect spacing is covered here. Perhaps a minor re-organization would put this connected material together in a single paragraph.

Line 92 – Please clarify the stocking. The ‘1 cow per 2-4 ha’ corresponds to 1 cow-calf pair, or a single animal (i.e., does a cow-calf pair = 2 cows in this situation)?

Line 116 – Not necessary to say ‘ are described below’.

Line 118 – Please define ‘T1’

Line 125-128 – Sentence describes two methods. Make separate sentences.

Line 128-130 – Please elaborate on the method for quantifying cover. What technique was used?

Line 135-136 = ‘When data … were available’

Line 141 – This is a really nice, super clear data analysis section!

Line 149 – Not familiar with the Tweedie – maybe provide a reference for this approach to your zero-inflated data.

Line 267-237 – I’m not understanding this argument that declines in native richness are greater in the ungrazed because they had low richness to begin with. Maybe expand on this thinking.

Line 254 – You should do this again – its only gotten worse in CA since 2017…

Line 275 – better word choice for ‘slackening’?

Line 298 – The thing I think this paper is missing – but not something you would address at this stage – is a community analysis because you have clearly observed a community shift toward more drought-tolerant species. Perhaps for the paper with 3 sampling periods in a few years!

7. PLOS authors have the option to publish the peer review history of their article (what does this mean?). If published, this will include your full peer review and any attached files.

Reviewer #3: **Yes: **Alejandro Huertas-Herrera

Reviewer #4: No

---

## [Author Response · Author response to Decision Letter 1]

27 Sep 2022

Please see the "Response to Reviewers" file which provides detailed responses to each of the reviewer comments.

---

## [Decision Letter · Decision Letter 2]

21 Nov 2022

Native annual forbs decline in California coastal prairies over 15 years despite grazing

PONE-D-22-07590R2

Dear Dr. Holl,

We’re pleased to inform you that your manuscript has been judged scientifically suitable for publication and will be formally accepted for publication once it meets all outstanding technical requirements.

Kind regards,

Mehdi Heydari

Academic Editor

PLOS ONE

Additional Editor Comments (optional):

Reviewers' comments:

Reviewer's Responses to Questions

**Comments to the Author**

1. If the authors have adequately addressed your comments raised in a previous round of review and you feel that this manuscript is now acceptable for publication, you may indicate that here to bypass the “Comments to the Author” section, enter your conflict of interest statement in the “Confidential to Editor” section, and submit your "Accept" recommendation.

Reviewer #3: All comments have been addressed

Reviewer #4: All comments have been addressed

2. Is the manuscript technically sound, and do the data support the conclusions?

Reviewer #3: Yes

Reviewer #4: Yes

3. Has the statistical analysis been performed appropriately and rigorously? 

Reviewer #3: Yes

Reviewer #4: Yes

4. Have the authors made all data underlying the findings in their manuscript fully available?

Reviewer #3: Yes

Reviewer #4: Yes

5. Is the manuscript presented in an intelligible fashion and written in standard English?

Reviewer #3: Yes

Reviewer #4: Yes

6. Review Comments to the Author

Reviewer #3: Dear authors. It is a pleasure to read your manuscript. Thank you for considering my suggestions. Congratulations for your manuscript! Best.

Reviewer #4: This manuscript has been reviewed the authors notified of the manuscript being acceptable. Therefore I have no additional comments.

7. PLOS authors have the option to publish the peer review history of their article (what does this mean?). If published, this will include your full peer review and any attached files.

Reviewer #3: **Yes: **Alejandro Huertas Herrera

Reviewer #4: No

---

## [Editor Report · Acceptance letter]

23 Nov 2022

PONE-D-22-07590R2 

Native annual forbs decline in California coastal prairies over 15 years despite grazing 

Dear Dr. Holl:

I'm pleased to inform you that your manuscript has been deemed suitable for publication in PLOS ONE. Congratulations! Your manuscript is now with our production department. 

Kind regards, 

on behalf of

Dr. Mehdi Heydari 

Academic Editor

PLOS ONE